# Causal Navigation by Continuous-time Neural Networks

**Charles Vorbach**[1,*], **Ramin Hasani**[1,*], **Alexander Amini**[1], **Mathias Lechner**[2], **Daniela Rus**[1]

## Abstract

Imitation learning enables high-fidelity, vision-based learning of policies within rich, photorealistic environments. However, such techniques often rely on traditional discrete-time neural models and face difficulties in generalizing to domain shifts by failing to account for the causal relationships between the agent and the environment. In this paper, we propose a theoretical and experimental framework for learning causal representations using continuous-time neural networks, specifically over their discrete-time counterparts. We evaluate our method in the context of visual-control learning of drones over a series of complex tasks, ranging from short- and long-term navigation, to chasing static and dynamic objects through photorealistic environments. Our results demonstrate that causal continuous-time deep models can perform robust navigation tasks, where advanced recurrent models fail. These models learn complex causal control representations directly from raw visual inputs and scale to solve a variety of tasks using imitation learning.

## 1 Introduction

Unlike machine learning systems, natural learning systems excel at generalizing learned skills beyond the original data distribution (Hasani et al., 2020, Hassabis et al., 2017, Sarma et al., 2018). This is due to the mechanisms they deploy during their learning process, such as active use of closed-loop interventions, accounting for distribution shifts, and the temporal structures (de Haan et al., 2019, Schölkopf, 2019). These factors are largely disregarded or are engineered away in the development of modern ML systems.

To take the first steps towards resolving these issues, we can investigate the spectrum of causal modeling (Peters et al., 2017). At one end of the spectrum, there are physical models of agents

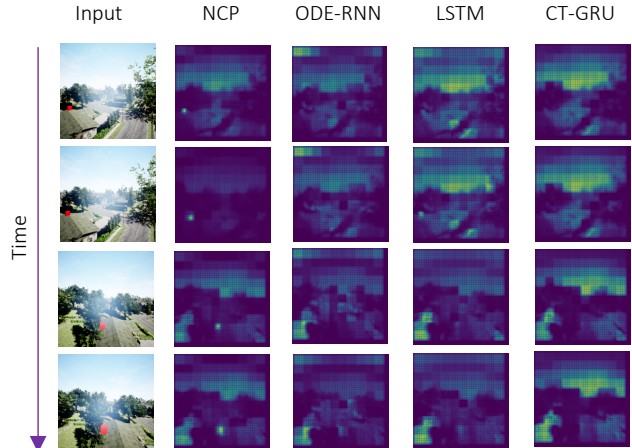

Figure 1: **Causal navigation from raw visual inputs**. Given a sequence of raw RGB inputs (left) a drone is trained to navigate towards the red-cube target. We visualize the saliency maps (right) for each model. Neural circuit policies (Lechner et al., 2020a) (a specific representation of CT-RNNs) can learn causal relationships (i.e., attend to the red-cube) directly from data while other models fail to do so. ODE-RNNs (Rubanova et al., 2019b), LSTM (Hochreiter and Schmidhuber, 1997) and CT- Gated Recurrent Units (Mozer et al., 2017). Saliency maps are computed by the visual backprop algorithm (Bojarski et al., 2016).

---

*Equal Contributions. [1]CSAIL MIT, [2]IST Austria, Correspondence to: rhasani@mit.edu Code and data are available at: https://github.com/mit-drl/deepdrone

35th Conference on Neural Information Processing Systems (NeurIPS 2021).

and environments described by differential equations. These physical models allow us to investigate interventions, predict future events using past information, and can describe the statistical dependencies in the system.

On the other end of the spectrum, statistical models allow us to construct dependencies and make predictions given independent and identically distributed (i.i.d.) (see Fig. 2). While physical models provide complete descriptions, it is intractable to define the differential equation systems that effectively model high-dimensional and complex sensory data. For long, causal modeling frameworks aims at bridging this gap to extract statistical dependencies while constructing causal graphs (Pearl, 2009, Spirtes et al., 2000) or structural causal models(Schölkopf, 2019) to intervene and explain relationships.

In this paper, we aim to use continuous-time (CT) neural networks (Chen et al., 2018b, Funahashi and Nakamura, 1993, Hasani et al., 2021b) equipped with

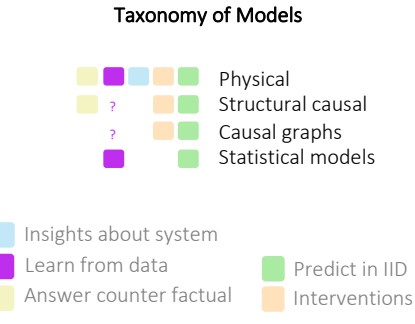

Figure 2: Taxonomy of models in the spectrum of causality. Figure data is taken from (Peters et al., 2017).

causal structures in order to get closer to the properties of physical models. In particular, we look into different representations of CT networks to see under what conditions and formulation they can form a causal model. We discover that the class of liquid time-constant networks (LTCs) (Hasani et al., 2021b) which are expressive continuous-time models constructed by bilinear approximation of neural ordinary differential equations (Chen et al., 2018b), satisfy the properties of a causal model. The uniqueness of their solution and their ability to capture both internal and external interventions make their forward- and backward- mode causal. Therefore, they can impose inductive biases on their architectures to learn causal representations.

We analyze how certain continuous-time (CT) neural networks are dynamical causal models. Furthermore, to justify the theoretical results, we empirically validate that these properties scale to high-dimensional visual input data in complex learning environments. We propose a series of control and navigation tasks for an end-to-end autonomous flight across various levels of complexity and temporal reasoning. We observe that traditional deep-learning models are capable of solving this task on offline, passive datasets but fail when deployed in closed-loop, active testing settings (de Haan et al., 2019, Wen et al., 2020). On the other hand, we find that only LTC-based models are able to complete the tasks in closed-loop interaction with the environments.

Specifically, consider a visual navigation task wherein a drone agent should learn to fly from point A to a target fixated at a point B, given only a sequence of raw visual inputs. The mapping between incoming pixels to the temporal and physical structure of the navigation is causal if we can explain and intervene at each step to observe how navigation decisions are decided based on the input pixels. We observe that in a neighborhood environment, agents based on LTCs (such as Neural Circuit Policies (Lechner et al., 2020a)) learn to stably (Lechner et al., 2020b) attend to the target at point B throughout the task's time horizon (Fig. 1). Therefore, *causes* of navigation to point B at a next time step is the agent's learned attention profile on the input images. These causal mappings are not present in other deep models (Fig. 1). Here, we show how LTCs' ability to capture causal structures directly from visual inputs results in improved robustness as well as interpretable decision making.

**Summary of Contributions.** i) We show theoretical evidence for the capability of CT neural networks in learning causal structures; ii) we perform extensive experiments supporting the effectiveness of causal CT models in visual drone navigation tasks with varying memory-horizons; and iii) we conduct robustness analysis of CT models in closed loop-testing within real-world scenarios.

## 2 Related Works

In this section, we describe research works that are closely related to the core findings of the paper.

**Causal Learning –** The dominant approach toward learning causal models are graphical methods (Ruggeri et al., 2007, Russell and Norvig, 2002), which try to model cause-effect relationships as a directed graph (Pearl, 2009, Pearl et al., 2009). Bayesian networks further combine graphical models

with Bayesian methods (Kemp et al., 2010, Ruggeri et al., 2007) to decompose the learning problem into a set of Bayesian inference sub-problems. (Weichwald et al., 2020) showed the effectiveness of such an approach for learning causal structures in nonlinear time-series via a set of linear models. Continuous-time Bayesian networks further adapted the idea to modeling cause-effect relationships in time-continuous processes (Gopalratnam et al., 2005, Nodelman et al., 2002, 2003, Nodelman, 2007). A different approach for causal modeling of time-continuous processes is to learn ODEs, which under certain conditions imply a structured causal model (Rubenstein et al., 2016). zIn this work, we describe a class of continuous models that has the ability to account for interventions and therefore captures the causal structures from data.

**Continuous-time Models –** CT models manifest a large range of benefits compared to discretized deep models. They can perform adaptive computations through continuous vector fields realized by advanced ODE solvers (Chen et al., 2018b). They are strong in modeling time-series data and realize memory and parameter efficiency (Chen et al., 2018b, Lechner et al., 2020a). A large number of alternative approaches have tried to improve and stabilize their training, namely the adjoint method (Gholami et al., 2019), use neural ODEs in complex time-series prediction tasks (Hasani et al., 2021a, Lechner and Hasani, 2020, Lechner et al., 2019, Rubanova et al., 2019b), characterize them better for inference and density estimation tasks (Dupont et al., 2019, Durkan et al., 2019, Hanshu et al., 2020, Holl et al., 2020, Jia and Benson, 2019, Liebenwein et al., 2021, Massaroli et al., 2020, Quaglino et al., 2020), and to verify them (Gruenbacher et al., 2021, Grunbacher et al., 2021). In this work, we prove an important property of the CT network: We show that a bilinear approximation of Neural ODEs can give rise to expressive causal models.

**Imitation Learning (IL) –** Imitation learning describes the task of learning an observation-action mapping from human demonstrations (Schaal, 1999). This objective can either be achieved via behavior cloning, which directly learns from observation-action pairs (Lechner et al., 2019, 2021), or indirectly via inverse reinforcement learning (Brunnbauer et al., 2021, Ng and Russell, 2000) which first constructs a reward function from an optimal policy. The most dominant behavior cloning paradigm is based on the DAgger framework (Ross et al., 2011), which iterates over the steps of expert data collection, supervised learning, and cloned policy evaluation. State-aware IL (Schroecker and Isbell, 2017) further adds a secondary objective to the learning task to bias the policy towards states where more training data is available. Recently, IL methods have been adapted to domains where the environment dynamics of the expert and learned policy mismatch (Desai et al., 2020).

More recent imitation learning one-shot methods pre-train policies via meta-learning to adapt to a task, such that task-specific behavior can be cloned with as little as a single demonstration (Duan et al., 2017, Yu et al., 2018). Alternatively, generative adversarial imitation learning phrases the behavior cloning problem as a min-max optimization problem between a generator policy and discriminator classifier (Ho and Ermon, 2016). Baram et al. (2017) extended the method by making the human expert policy end-to-end differentiable. The method has been further adapted to imperfect (Wu et al., 2019) and incomplete demonstrations (Sun and Ma, 2019).

**Visual Navigation –** Cognitive mapping and planning (Gupta et al., 2017a,b) addresses the problem of learning to navigate from visual input streams by constructing a map of the environment and plan the agent's actions to achieve a given goal. Chen et al. (2018a) adapted the approach for the goal of exploring and mapping the environment. An alternative method represents the map in the form of a graph (Savinov et al., 2018). Neural SLAM (simultaneous location and mapping) shares many characteristics of cognitive mapping and planning (Chaplot et al., 2019) but takes one step further in separating mapping, pose estimation, and goal-oriented navigation. Target driven navigation, where the agent must find a target in an unknown environment, has seen successes using both semantic segmentation (Mousavian et al., 2019), and neural SLAM (Chaplot et al., 2020).

Moreover, visual navigation for learning-to-drive context, have extensively studied the causal confusion problem (de Haan et al., 2019), and the generalization of the imitation learning problems, through using modules to extract useful priors from pixel inputs (Chen et al., 2020, Filos et al., 2020, Park et al., 2020, Rhinehart et al., 2019, Sauer et al., 2018). These methods can benefit from the LTC-based networks designed in our present study, to enhance their knowledge distillation pipelines.

# 3   Problem Setup

In this section, we describe necessary concepts to formally derive the main results of our report.

## 3.1 Causal Structures

In a structural causal model (SCM), given a set of observable random variables $X_1, X_2, \ldots, X_n$, as vertices of a directed acyclic graph (DAG), we can compute each variable from the following assignment (Schölkopf, 2019):

$$X_i := f_i(\mathbf{PA}_i, U_i), \quad i = 1, \ldots, n. \tag{1}$$

Here, $f_i$ is a deterministic function of the parents of the event, $X_i$, in the graph ($\mathbf{PA}_i$) and of the stochastic variable, $U_i$. One can intuitively think of the causal structure framework as a function estimation problem rather than in terms of probability distributions (Spirtes et al., 2000). Direct causation is implied by direct edges in the graph through the assignment described in Eq. 1. The stochastic variables $U_1, \ldots, U_n$ ensure that a joint distribution $P(X_i | \mathbf{PA}_i)$ is constructed as a general objective (Pearl, 2014, Schölkopf, 2019).

The SCM framework enables us to explore through the known physical mechanisms and functions to build flexible probabilistic models with *interventions*, replacing the "slippery epistemic probabilities, $P(X_i, \mathbf{PA}_i)$, with which we had been working so long" in the study of machine learning systems (Pearl, 2009, Schölkopf, 2019). *Interventions* can be formalized by the SCM framework, as operations that alter a subset of properties of Eq. 1. For instance, modifying $U_i$, or replacing $f_i$ (and as a result $X_i$) (Karimi et al., 2020). Moreover, by assuming joint independence of $U_i$s, we can construct *causal* conditionals known as causal (disentangled) factorization, as follows (Schölkopf, 2019):

$$p(X_1, \ldots, X_n) = \prod_{i=1}^{n} p(X_i | \mathbf{PA}_i). \tag{2}$$

Eq. 2 stands for the causal mechanisms by which we can model all statistical dependencies of given observables. Accordingly, Causal learning involves the identification of the causal conditionals, the function $f_i$, and the distribution of $U_i$s in assignment Eq. 1.

## 3.2 Differential Equations Can form Causal Structures

Physical dynamics can be modeled by a set of differential equations (DEs). DEs allow us to predict the future evolution of a dynamical system and describe its behavior as a result of interventions. Their coupled time-evolution enables us to define averaging mechanisms for computing statistical dependencies (Peters et al., 2017). A system of differential equations enhances our understanding of the underlying physical phenomenon, explains its behavior, and dissects its causal structure.

For instance, consider the following system of DEs: $\frac{d\mathbf{x}}{dt} = g(\mathbf{x})$, $\mathbf{x} \in \mathbb{R}^d$, with initial values at $x_0$, where $g$ is a nonlinear function. The Picard-Lindelöf theorem (Nevanlinna, 1989) states that a differential equation of the form above would have a unique solution as long as $g$ is Lipschitz. Therefore, if we unroll the system to infinitesimal differentials using the explicit Euler method, we get: $\mathbf{x}(t + \delta t) = \mathbf{x}(t) + dt f(\mathbf{x})$. This representation under the uniqueness condition shows that the near future events of $x$ are predicted using its past information, thus, forming a causal structure.

Thus, a DE system is a causal structure that allows us to process the effect of interventions on the system. On the other side of the spectrum of causal modeling (Peters et al., 2017), pure statistical models allow us to learn structures from data with little insight about causation and associations between epiphenomena. Since causality aims to bridge this gap, in this paper, we propose to construct causal models with continuous-time neural networks.

## 3.3 Continuous-time Neural Networks

CT networks are a class of deep learning models with their hidden states being represented by ordinary differential equations (ODEs) (Funahashi and Nakamura, 1993). The hidden state $x(t)$ of a neural network $f$ is computed by the solution of the initial value problem below (Chen et al., 2018b):

$$\frac{dx}{dt} = f(\mathbf{x}(t), t, \theta), \quad \mathbf{x} \in \mathbb{R}^d, \tag{3}$$

where $f$ is parametrized by $\theta$. CT models enable approximation of a class of functions which we did not know how to generate otherwise (Chen et al., 2018b, Hasani et al., 2021b). They can be used in both inference and density estimation with constructing a continuous flow to model data,

efficiently (Dupont et al., 2019, Grathwohl et al., 2018, Rubanova et al., 2019a). CT models can be formulated in different representations. For instance, to achieve stability, CT recurrent neural networks (CT-RNNs) were introduced in the following form (Funahashi and Nakamura, 1993):

$$\frac{d\mathbf{x}(t)}{dt} = -\frac{\mathbf{x}(t)}{\tau} + f(\mathbf{x}(t), t, \theta), \tag{4}$$

where the term $-\frac{\mathbf{x}(t)}{\tau}$ derives the system to equilibrium with a time-constant $\tau$. To increase expressivity (Raghu et al., 2017), liquid time-constant networks (LTCs) with the following representation can be used (Hasani et al., 2021b):

$$\frac{d\mathbf{x}(t)}{dt} = -\left[\frac{1}{\tau} + f(\mathbf{x}(t), \mathbf{I}(t), t, \theta)\right] \odot \mathbf{x}(t) + f(\mathbf{x}(t), \mathbf{I}(t), t, \theta) \odot A. \tag{5}$$

In Eq. 5, $\mathbf{x}^{(D \times 1)}(t)$ is the hidden state of an LTC layer with D cells, $\mathbf{I}^{(m \times 1)}(t)$ is the input to the system, $\tau^{(D \times 1)}$ is the fixed internal time-constant vector), $A^{(D \times 1)}$ is an output control bias vector, and $\odot$ is the Hadamard product. Here, the ODE system follows a bilinear dynamical system (Penny et al., 2005) approximation of Eq. 3 to construct an input-dependent nonlinearity in the time-constant of the differential equation. This was shown to significantly enhance the expressive power of CT models in robotics and time-series prediction tasks (Hasani et al., 2021b, Lechner et al., 2020a). In the following, we show how CT models can be designed as causal structures to perform more interpretable real-world applications.

## 4 Results

In this section, we first explain how a continuous-time model identified by Eq. 3, standalone, cannot satisfy the causal structure properties (Peters et al., 2017) even under Lipschitzness of its network. We then show that the class of liquid time-constant networks can enable causal modeling.

### 4.1 Causal Modeling with Continuous-time Networks

**Neural ODEs cannot account for external interventions.** Let $f$ be the nonlinearity of a continuous-time neural network. Then the learning system defined by Eq. 3, form temporal causation, however cannot account for external interventions (change of the environment conditions), and therefore does not form a causal structure even when $f$ is Lipschitz-continuous.

To describe this in detail, we unfold the ODE by infinitesimal differentials as follows:

$$\mathbf{x}(t + \delta t) = \mathbf{x}(t) + dt f(\mathbf{x}, \theta). \tag{6}$$

If $f$ is Lipschitz continuous, based on Picard-Lindelöf's existence theorem, the trajectories of this ODE system are unique and thus invertible. This means that, at least locally, the future events of the system can be predicted by its past values. Since the transformation is invertible, this setting is also true if we run the ODE backward in time (e.g., during the training process).

When the ODE system is trained by maximum likelihood estimation, given an initial weight distribution, the statistical dependencies between the system's variables might emerge from data. The resulting statistical model can predict in i.i.d. setting and learn from data, but it cannot predict under distribution shift or implicit/explicit interventions, simply because the system's semantics does not have input-dependent terms. Thus, the system cannot answer counterfactual questions (Mooij et al., 2013).[2] Although Neural ODEs in their generic representation cannot account for interventions, their other forms can help design a causal model.

**LTCs are Dynamic Causal Models.** LTCs described by Eq. 5 resemble the representation of a simple Dynamic Causal Model (DCM) (Friston et al., 2003) with a bilinear Taylor approximation (Penny et al., 2005). DCMs aim to extracting the causal architecture of a given dynamical systems. DCMs represent the dynamics of the hidden nodes of a graphical model by ODEs, and allow for feedback connectivity structures unlike Bayesian Networks (Friston et al., 2003). A simple DCM

---

[2] A counterfactual question describe a causal relationship of the form: "If X had not occurred, Y would not have occurred (Molnar, 2020)"

can be designed by a second-order approximation (bilinear) of a given system such as $d\mathbf{x}/dt = F(\mathbf{x}(t), \mathbf{I}(t), \theta)$, as follows (Friston et al., 2003):

$$dx/dt = (A + \mathbf{I}(t)B)\mathbf{x}(t) + C\mathbf{I}(t) \tag{7}$$

$$A = \frac{\partial F}{\partial \mathbf{x}(t)}\Big|_{I=0}, \quad B = \frac{\partial^2 F}{\partial \mathbf{x}(t)\partial \mathbf{I}(t)}, \quad C = \frac{\partial F}{\partial \mathbf{I}(t)}\Big|_{x=0},$$

where $\mathbf{I}(t)$ is the inputs to the system, and $\mathbf{x}(t)$ is the nodes' hidden states. A key property of a DCM representation is its ability to capture both internal and external causes on the system (interventions). Matrix A is a fixed internal coupling of the system. Matrix B controls the impact of the inputs on the coupling sensitivity among the network's nodes (controlling internal interventions). Matrix C embodies the external inputs' influence on the state of the system (controlling external interventions).

DCMs have shown promise in learning the causal structure of the brain regions in complex time-series data (Breakspear, 2017, Ju and Bassett, 2020, Penny et al., 2005). DCMs can be extended to a universal framework for causal function approximation by neural networks through the LTC neural representations (Hasani et al., 2021b).

**Proposition 1.** *Let $f$ be the nonlinearity of a given LTC network identified by Eq. 5. Then the learning system identified by Eq. 5 can account for internal and external interventions by its weight parameters $\theta = \{W_r^{(D \times D)}, W^{(D \times m)}, b^{(D \times 1)}\}$, for D LTC cells, and input size $m$ and $A^{(D \times 1)}$, and therefore, forms a dynamical causal model, if $f$ is Lipschitz-continuous.*

*Proof.* To prove this, we need to show two properties for the LTC networks: 1) Uniqueness of their solution 2) Existence of internal and external intervention coefficients:

*Uniqueness.* By using the Picard-Lindelöf theorem (Nevanlinna, 1989), it was previously shown that for an $F : \mathbb{R}^D \to \mathbb{R}^D$, bounded $C^1$-mapping, the differential equation:

$$\dot{x} = -(1/\tau + F(x))x + AF(x),$$

has a unique solution on $[0, \infty)$. (Full proof in Lemma 5 of (Hasani et al., 2021b)).

*Interventions Coefficients.* Let $f$ be a Lipschitz-continuous activation function such as tanh, then $f(\mathbf{x}(t), \mathbf{I}(t), t, \theta) = tanh(W_r\mathbf{x} + W\mathbf{I} + b)$. If we set $x = 0$, in Eq. 5, then the external intervention coefficients $C$ in Eq. 7 for LTCs can be obtained by:

$$\frac{\partial F}{\partial \mathbf{I}}\Big|_{x=0} = W(1 - f^2) \odot A$$

The corresponding internal intervention coefficients $B$ of Eq. 7, for LTC networks becomes:

$$\frac{\partial^2 F}{\partial \mathbf{x}(t)\partial \mathbf{I}(t)} = W(f^2 - 1) \odot \big[2W_r f \odot (A - x) + 1\big]$$

This shows that by manipulating matrices $W$, $W_r$, and $A$ one can control internal and external interventions to an LTC system which gives the statement of the proposition. $\square$

Proposition 1 shows that LTCs are causal models in their forward pass, and by manipulation of their parameters, one can gain insights into the underlying systems. We next show that LTCs' backward path also gives rise to a causal model. This implies that LTCs trained from demonstration via reverse-mode automatic differentiation (Rumelhart et al., 1986) can give rise to a causal model.

## 4.2 Training LTCs via Gradient Descent Yields Causal Models

The uniqueness of the solution of LTCs allows any ODE solver to reproduce a forward pass trajectory with backward computations from the end point of the forward pass. This property enables us to use the backpropagation algorithm, either by using the adjoint sensitivity (Pontryagin, 2018) method or backpropagation through time (BPTT) to train an LTC system. Formally, for a forward pass trajectory of the system states $\mathbf{x}(t)$ of an LTC from $t_0$ to $t_n$, we can compute a scalar value loss, $L$ at $t_n$ by:

$$L(\mathbf{x}(t_n)) = L\Big(\mathbf{x}(t_0) + \int_{t_0}^{t_n} \frac{d\mathbf{x}}{dt}dt\Big). \tag{8}$$

Suppose we use the adjoint method for training the network, then the augmented state is computed by $a(t) = \frac{\partial L}{\partial \mathbf{x}(t)}$, whose kinetics are determined by $\frac{da}{dt} = -a^T(t)\frac{\partial LTC_{rhs}}{\partial \mathbf{x(t)}}$ (Chen et al., 2018b). To compute the loss gradients in respect to the parameters, we run the adjoint dynamics backwards from gradients $\frac{dL}{d\mathbf{x}(t_n)}$ and states $\mathbf{x}(t_n)$, while solving a third integral in reverse as follows: $\frac{dL}{d\theta} = -\int_{t_n}^{t_0} a^T(t)\frac{\partial LTC_{rhs}}{\partial \theta}dt$. All integrals can be solved in reverse-mode by a call to the ODE solver.

Based on Proposition 1, for an LTC network at each training iteration, the internal and external interventions not only help the system learn statistical dependencies from data but also facilitate the learning of the causal mapping. These results bridge the gap between pure physical models and causal structural models to obtain better learning systems.

**Real-world implications.** Consider a simple drone navigation task in which the objective is for the drone to navigate from position A to a target position B while simultaneously avoiding obstacles. In this setting, the agent is asked to infer high-level control signals directly from visual inputs to reach the target goal. Even the simplest vision-based deep model can accomplish this task in passive open-loop settings (Lechner et al., 2020a). However, completion in a closed-loop environment while inferring the true causal structure of the task is significantly more challenging. For instance, where should the agent attend to for taking the next action? With what mechanisms does the system infer the objective of the task? And accordingly, how robust are the decisions of the learned policy?

In the next section, we perform an extensive experimental evaluation to find answers to the above questions. We also aim to validate our theoretical results on the capability of LTC models to capture the true causal structure of a high-dimensional visual reasoning task, where other models fail.

## 5 Experiments

We designed photorealistic visual navigation tasks with varying memory horizons including (1) navigating to a static target, (2) chasing a moving target, and (3) hiking with guide markers (Fig. 3).

**Experimental setup.** We designed tasks in Microsoft's AirSim (Madaan et al., 2020) and Unreal Engine. To create data for imitation learning, we use greedy path search with a Euclidean heuristic over unoccupied voxels to obtain knot points for cubic spline interpolation, which is then followed via a pure pursuit controller. This strategy is modified for each task (details in the supplements).

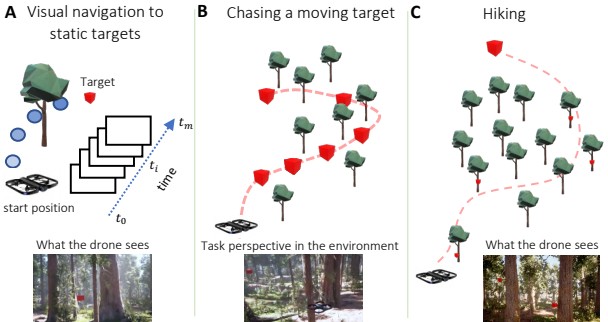

Figure 3: Visual drone navigation tasks. A) Navigation to a static target, B) Chasing a moving target, C) Hiking with a set of markers in the environment

**Baselines.** We evaluate NCP networks (Lechner et al., 2020a) against a set of baseline models. This includes ODE-RNNs (Rubanova et al., 2019b) which are the recurrent network version of Neural ODEs (Chen et al., 2018b), long short-term memory networks (LSTMs) (Hochreiter and Schmidhuber, 1997), and CT-GRU networks (Mozer et al., 2017), which are the continuous equivalent of GRUs (Chung et al., 2014). These baselines are chosen to validate our theoretical results: Not all CT models are causal models. Similarly, discretized RNN models, such as LSTMs, perform well on tasks with long-term dependencies, yet they are not causal models. In Section 4, we showed that NCPs, which are sparse neural networks built based on LTC neurons (Hasani et al., 2021b), are dynamical causal models. Therefore, they can learn the true causation of a given task. In our experiments, camera images are perceived by convolutional layers and are fed into the RNN networks which act as a controller. For a fair comparison, the number of trainable parameters of all models is within the same range, and they are all trained by Adam optimizer (Kingma and Ba, 2014) with a cosine similarity loss (See more details in the supplements).

### 5.1 Navigation to Static Target with Occlusion

In this task, the drone navigates to a target marker that is less than 25 meters away and visible to it. We place a red cube on a random unoccupied voxel in the environment to function as the target

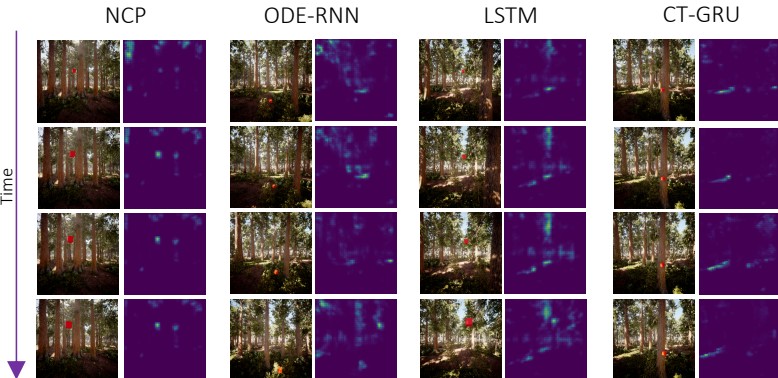

Figure 4: Navigation to a static target in closed-loop environments. NCPs are the only models that can capture the causal structure of the tasks directly from visual data.

marker. We constrain the target marker to appear in the drone's viewing frustum for most cases. Though there may be occlusion of the target upon random respond of the drone in the environment (more details in the supplements). Occlusions create temporal dependencies. We tested static CNN networks and observed that they fail to perform scenarios in which the target is occluded. Table 1 shows that in both neighborhood and forest environments, all agents learn the task with a reasonable validation loss in a passive imitation learning setting. However, once these agents are deployed with closed-loop control, we observed that the success rates for LSTM and ODE-RNNs drop to only 24% and 18% of 50 attempted runs. CT-GRU managed to complete this task in 40% of the runs, whereas NCPs completed the tasks in 48% of the runs. Why is this the case?

To understand these results better, we used the Visual-Backprop algorithm (Bojarski et al., 2016) to compute the saliency maps of the learned features in the input space. Saliency maps would show us where the attention of the network was when taking the next navigation decision. As shown in Fig. 4, we observe that NCP has learned to attend to the static target within its field of view to make a future decision. This attention profile was not present in the saliency maps of the other agents. LSTM agents are sensitive to lighting conditions compared to the CT models. This experiment supports our theoretical results on the ability of LTC-based models to learn causal representations.

Table 1: Validation on short-term navigation. Cosine similarity loss [-1, 1] (smaller is better), n=5.

| Algorithms | Environments | |
|---|---|---|
| | RedWood Forest | Neighborhood |
| LSTM | $-0.823 \pm 0.006$ | $-0.838 \pm 0.019$ |
| ODE-RNN | $-0.815 \pm 0.043$ | $-0.855 \pm 0.019$ |
| CT-GRU | $-0.855 \pm 0.001$ | $\mathbf{-0.877 \pm 0.002}$ |
| NCP (ours) | $\mathbf{-0.859 \pm 0.035}$ | $-0.855 \pm 0.008$ |

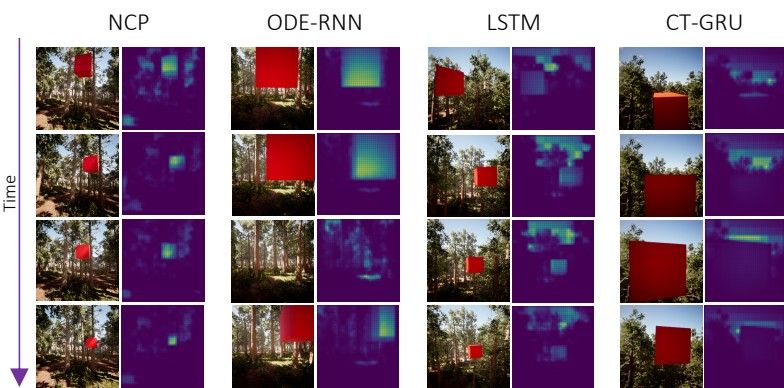

Figure 5: Chasing a moving target in closed-loop environments. NCPs are the only models that can capture the causal structure of the tasks directly from visual data.

Table 2: Chasing objects, Validation performance with co-sine similarity loss (smaller is better), n=5

Table 3: Hiking, Validation performance with co-sine similarity loss (smaller is better), n=5

| Algorithms | Environments | | Algorithms | Environments | |
|---|---|---|---|---|---|
| | RedWood Forest | Neighborhood | | RedWood Forest | Neighborhood |
| LSTM | -0.943 ± 0.028 | -0.947 ± 0.008 | LSTM | -0.273 ± 0.388 | **-0.781 ± 0.030** |
| ODE-RNN | **-0.967 ± 0.009** | -0.953 ± 0.011 | ODE-RNN | **-0.896 ± 0.026** | -0.710 ± 0.003 |
| CT-GRU | -0.958 ± 0.017 | **-0.979 ± 0.003** | CT-GRU | -0.359 ± 0.073 | -0.725 ± 0.086 |
| NCP (ours) | -0.936 ± 0.022 | **-0.975 ± 0.012** | NCP (ours) | -0.676 ± 0.192 | -0.711 ± 0.013 |

## 5.2 Chasing a Moving Target

In this task, the drone follows a target marker along a smooth spline path. Using a generate and test method, we create a path for the drone to follow by using a random walk with momentum to find knot points for fitting a spline (details in the supplements). Table 2 shows that all agents were able to learn to follow their targets in a passive open-loop case. However, similar to the previous experiment, we witnessed that not all models can successfully complete the task in a closed-loop setting where interventions play a big role. NCPs were 78% successful at completing their task, while LSTM in 66%, ODE-RNNs in 52%, and CT-GRU in 38% were successful. Once again, we looked into the attention maps of the models, illustrated in Fig. 5. We see that CT-GRU networks did not learn to attend to the target they follow. LSTMs again show sensitivity to lighting conditions. ODE-RNNs keep a close distance to the target, but they occasionally lose the target. In contrast, NCPs have learned to attend to the target and follow them as they move in the environment.

## 5.3 Hiking Through an Environment

In this task, the drone follows multiple target markers which are placed on the surface of obstacles within the environment (Fig. 3C) (Experimental details in Supplements). This task is significantly more complex than the previous tasks, especially when agents are deployed directly in the environment. This is because the agents have to learn to follow a much longer time-horizon task, by visual cues, in a hierarchical fashion.

Interestingly, we see most agents learn a reasonable degree of validation loss during the learning process as depicted by Table 3. Even ODE-RNNs realize excellent performance in the passive setting. However, when deployed in the environment, none of the models other than NCP could perform the task completely in 50 runs. NCPs could perform 30% successfully thanks to their causal structure.

**How to improve the performance of models in the hiking task?** In order to improve the performance on a purely visually-navigated hiking task, an agent must be supplied with a long-term memory component (much longer than that of LSTMs). All ODE-based models including NCPs require a gradient wrapper to be able to perform well on tasks with very long-term dependencies. For instance, mixed memory architectures such as ODE-LSTM (Lechner and Hasani, 2020) can use NCPs as their continuous-time memory wrapped together with gradient propagation mechanisms enabled by an LSTM network, to perform better on the hiking task.

**CNN network's performance in all scenarios.** In Table 4, we summarized the success rate of CNNs in all tasks when deployed in closed-loop. As expected, we observe that when temporal dependencies in the tasks appear (such as Occlusion, or Hiking) as well as when the input images are highly perturbed (such as heavy rain and Fog) the performance of CNNs drastically decreases. This observation validates that having a memory component, in all tasks is essential.

Table 4: Closed-loop evaluation of trained policies on various navigation and interaction tasks. Agents and policies are reinitialized randomly at the beginning of each trial (n=50). Values correspond to success rates (higher is better).

| Model | Static Target | | | | | Chasing | | | | Hiking |
|---|---|---|---|---|---|---|---|---|---|---|
| | Clear | Fog | Light Rain | Heavy Rain | Occlusion | Clear | Fog | Light Rain | Heavy Rain | Clear |
| **CNN** | 36% | 6% | 32% | 2% | 4% | 50% | 42% | 54% | 28% | 0% |
| **LSTM** | 24% | 22% | 22% | 4% | 20% | 66% | **62%** | 56% | 44% | 2% |
| **ODE-RNN** | 18% | 10% | 18% | 2% | 24% | 52% | 42% | 62% | 44% | 4% |
| **CT-GRU** | 40% | 8% | **60%** | 32% | 28% | 38% | 36% | 48% | 42% | 0% |
| **NCP (ours)** | **48%** | **40%** | 52% | **60%** | **32%** | **78%** | 52% | **84%** | **54%** | **30%** |

# 6  Discussions, Scope and Conclusions

We provided theoretical evidence for the capability of different representations of continuous-time neural networks in learning causal structures. We then performed a set of experiments to confirm the effectiveness of continuous-time causal models in high-dimensional and visual drone navigation tasks with different memory-horizons compared to other methods. We conclude that the class of liquid time-constant networks has the great potential of learning causal structures in closed-loop reasoning tasks where other advanced RNN models cannot perform well.

**Model performances drop significantly in closed-loop.** Table 4 shows a summary of success episodes when the agents were deployed in various navigation tasks. As the memory-horizon and the complexity of the task increases, models that are not causal struggle with close-loop interactions with the environment. Continuous-time models with causal structures, such as NCPs, considerably improve the real-world performance of agents.

**Robustness to environmental perturbations.** We next investigated how robust are these models under heavy environmental perturbations such as rain. We used the AirSim weather API to include heavy rain and fog in closed-loop tests of the static target and chasing a moving target task (Fig. 6).

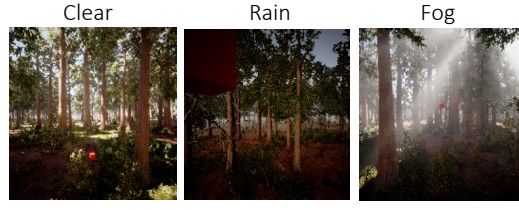

Figure 6: Sample input images from the forest environment. While agents are trained on offline, passive datasets from the Clear environment, testing is performed in an active control setup under various levels of environmental perturbations.

We observed that under high-intensity rain, all models have a performance drop, but NCPs show more resiliency to these perturbations. High-intensity rain and fog (Table 4) confuses LSTM and ODE RNNs the most where they are not able to complete the task.

**When shall we use temporal convolutions and attention-based methods?** Both temporal convolution (Bai et al., 2018) and attention-based architectures (Vaswani et al., 2017) are reasonable choices for many spatiotemporal tasks (Kaufmann et al., 2020, Lee et al., 2020). However, they require the user to explicitly set the temporal horizon of the observation memory, whereas RNNs learn the temporal memory component, implicitly. Consequently, RNNs are usually the preferred choice when the temporal horizon is not known or varies, which is the case in our experiments.

**Handling Occlusions and temporal dependencies.** All tasks discussed require temporal dependencies because of environmental line-of-sight occlusions between the drone and the target. We specifically observed that NCPs are capable of handling more complex runs (where occlusions are maximal) and other models fail. We have also tested these scenarios with (single frame) CNNs and observed that the model could not complete the task successfully in all cases where the marker is occluded. CNNs achieved only a 14% success rate in static target runs with an occlusion which is at least 50% lower than the results of the other recurrent models shown in Table 4.

**If temporal samples arrive at a constant rate, why a continuous-time model performs better?** The use of continuous models (implemented by ODEs) for continuous problems can potentially lead to the design of better causal mechanisms (getting closer to physical modeling in the taxonomy of causal models). Moreover, many recent works show advanced CT models can outperform advanced discretized RNNs even when the incoming samples are equidistant (Erichson et al., 2021, Rusch and Mishra, 2021). Also, CT-models realize a novel category of functions which was not possible to realize otherwise (i.e., the vector fields realized by complex ODE solvers cannot be achieved by discretized models) (Chen et al., 2018b).

**It is not trivial to show causal properties for gated RNNs.** In gated recurrent networks (e.g., LSTMs), the gating mechanisms can be counted as an implicit intervention to the state representation. However, a direct theoretical framework to prove this is not feasible as there is no direct relation between dynamic causal models and LSTMs, for instance. Besides, our empirical study in visual navigation tasks suggests that the gating mechanism does not help causality (See for instance the attention maps presented in Figures. 1, 4, and 5.)

In summary, we showed that special presentation of CT models realize causal models and can significantly enhance decision making in real-world applications.

## Acknowledgments

C.V., R.H. A.A. and D.R. are partially supported by Boeing and MIT. A.A. is supported by the National Science Foundation (NSF) Graduate Research Fellowship Program. M.L. is supported in part by the Austrian Science Fund (FWF) under grant Z211-N23 (Wittgenstein Award). Research was sponsored by the United States Air Force Research Laboratory and the United States Air Force Artificial Intelligence Accelerator and was accomplished under Cooperative Agreement Number FA8750-19-2-1000. The views and conclusions contained in this document are those of the authors and should not be interpreted as representing the official policies, either expressed or implied, of the United States Air Force or the U.S. Government. The U.S. Government is authorized to reproduce and distribute reprints for Government purposes notwithstanding any copyright notation herein.

## Funding Transparency Statement

Authors declare no competing interests. *Funding in direct support of this work:* The Boeing Company through the Explainable Control Project, National Science Foundation (NSF) Graduate Research Fellowship Program, Austrian Science Fund (FWF) under grant Z211-N23 (Wittgenstein Award), the United States Air Force Research Laboratory and the United States Air Force Artificial Intelligence Accelerator and was accomplished under Cooperative Agreement Number FA8750-19-2-1000.

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
