## S1 Code and Data Reproducability

All code and data are released in https://github.com/mit-drl/deepdrone

## S2 Compute Derivatives in the Proof of Proposition 1.

An LTC network has the following representation Hasani et al. (2021b):

$$\frac{d\mathbf{x}(t)}{dt} = -\left[\frac{1}{\tau} + f(\mathbf{x}(t), t, \theta)\right] \odot \mathbf{x}(t) + f(\mathbf{x}(t), t, \theta) \odot A.$$

In 5, $\mathbf{x}^{(D \times 1)}(t)$ is the hidden state of an LTC layer with D cells, $\tau^{(D \times 1)}$ is the fixed internal time-constant vector), $A^{(D \times 1)}$ is an output control bias vector, and $\odot$ is the Hadamard product. A simple DCM can be designed by a second-order approximation (bilinear) of a given system such as $d\mathbf{x}/dt = F(\mathbf{x}(t), \mathbf{I}(t), \theta)$, as follows Friston et al. (2003):

$$d\mathbf{x}/dt = (A + \mathbf{I}(t)B)\mathbf{x}(t) + C\mathbf{I}(t)$$

$$A = \frac{\partial F}{\partial \mathbf{x}(t)}\Big|_{I=0}, \quad B = \frac{\partial^2 F}{\partial \mathbf{x}(t) \partial \mathbf{I}(t)}, \quad C = \frac{\partial F}{\partial \mathbf{I}(t)}\Big|_{x=0},$$

*Interventions Coefficients.* Let $f$ be a Lipschitz-continuous activation function such as tanh, then $f(\mathbf{x}(t), \mathbf{I}(t), t, \theta) = tanh(W_r\mathbf{x} + W\mathbf{I} + b)$. If we set $x = 0$, in Equation 5, then the external intervention coefficients $C$ in (7) for LTCs can be obtained by:

$$\frac{\partial F}{\partial \mathbf{I}}\Big|_{x=0} = W(1 - f^2) \odot A$$

The corresponding internal intervention coefficients $B$ of Eq. 7, for LTC networks can be computed by the partial derivatives of the Eq. 5's right hand-side (rhs) $\frac{\partial^2 F}{\partial \mathbf{x}(t) \partial \mathbf{I}(t)}$.

We first compute the partials $\frac{\partial F}{\partial \mathbf{x}(t)}$ and $\frac{\partial F}{\partial \mathbf{I}(t)}$ and then $\frac{\partial^2 F}{\partial \mathbf{x}(t) \partial \mathbf{I}(t)}$

$$\begin{aligned}
\frac{\partial F}{\partial \mathbf{x}(t)} =& -\frac{1}{\tau} - \tanh(W_r\mathbf{x} + W\mathbf{I} + b) \\
& - W_r\Big(1 - \tanh^2(W_r\mathbf{x} + W\mathbf{I} + b)\Big)\mathbf{x} \\
& + W_r\Big(1 - \tanh^2(W_r\mathbf{x} + W\mathbf{I} + b)\Big)A \\
=& -\frac{1}{\tau} - \tanh(W_r\mathbf{x} + W\mathbf{I} + b) + W_r\Big(1 - \tanh^2(W_r\mathbf{x} + W\mathbf{I} + b)\Big)(A - \mathbf{x}) \\
=& -\frac{1}{\tau} - f + W_r(1 - f^2)(A - \mathbf{x})
\end{aligned} \tag{S1}$$

$$\begin{aligned}
\frac{\partial F}{\partial \mathbf{I}(t)} =& -W\Big(1 - \tanh^2(W_r\mathbf{x} + W\mathbf{I} + b)\Big)\mathbf{x} + W\Big(1 - \tanh^2(W_r\mathbf{x} + W\mathbf{I} + b)\Big)A \\
=& W\Big(1 - \tanh^2(W_r\mathbf{x} + W\mathbf{I} + b)\Big)(A - \mathbf{x}) \\
=& W(1 - f^2)(A - \mathbf{x})
\end{aligned} \tag{S2}$$

$$\begin{aligned}
\frac{\partial^2 F}{\partial \mathbf{x}(t) \partial \mathbf{I}(t)} =& \frac{\partial F}{\partial \mathbf{x}(t)}\Big(\frac{\partial F}{\partial \mathbf{I}(t)}\Big) = \\
& - 2WW_r \tanh(W_r\mathbf{x} + W\mathbf{I} + b)\Big(1 - \tanh^2(W_r\mathbf{x} + W\mathbf{I} + b)\Big)(A - \mathbf{x}) \\
& - W\Big(1 - \tanh^2(W_r\mathbf{x} + W\mathbf{I} + b)\Big) \\
=& W(f^2 - 1) \odot \Big[2W_r f \odot (A - x) + 1\Big]
\end{aligned} \tag{S3}$$

This shows that by manipulating matrices $W$, $W_r$, and $A$ one can control internal and external interventions to an LTC system which gives the statement of the proposition.

## S3 Experimental Setup.

In the following, we explain in detail the technical setup of our experiments for data collection and task design.

The voxel occupancy cache is produced using simulated Lidar data available from the AirSim APIs Madaan et al. (2020). During the control step, each point returned by the simulated Lidar is rounded to the nearest voxel and then placed inside of an LRU cache with a maximum length of 100,000 entries. A voxel size of 1 meter is used, and the simulated Lidar has a maximum range of 45 meters. This produces a voxel occupancy map of the local area around the drone, which is used for navigation.

Images from the drone's frontal camera and the drone's coordinate position are recorded at 20 Hz. The drone's yaw angle is controlled so that the frontal camera always faces the relevant target marker.

### S3.1 Task Design Setting

**Navigation to Static Target.** In this task, the drone navigates to a target marker that is less than 25 meters away and visible to it. We place a red cube on a random unoccupied voxel in the environment to function as the target marker. By constraining the target marker to appear in the drone's viewing frustum and to have an uninterrupted line of sight through the voxel occupancy cache, we minimize occlusions.

---

**Algorithm S1** Navigation Tasks

---

**Input:** set of occupied voxels, $O$
**while** $x \in O$ **do**
    $x = $ RANDOMVOXEL()
**end while**
NAVIGATETOENDPOINT($x$).

---

**Algorithm S2** NAVIGATETOENDPOINT($x$)

---

**Input:** endpoint, $x \in \mathbb{R}^3$
$s = $ PLANPATH($x$)
**while** $||$DRONEPOSITION() - $x$ $||$ **do**
    $s = $ PLANNINGTHREAD(PLANPATH, $x$)
    MOVETO(PURSUITPOINT(s))
**end while**

---

During data collection, we perform greedy path search over the unoccupied voxels, ignoring voxels from which the target marker is occluded or outside the viewing frustum. When a path is found, a cubic spline is interpolated between the voxel points. After this initial path is found, the control algorithm begins. Re-planning continues in a separate thread, since as the drone moves and the occupancy cache is updated, voxels in the initial path may be observed to be occupied.

The control algorithm used for data collection follows this spline with a pure pursuit algorithm, which is tuned for the drone's speed. If the planning thread returns a new spline, we update the control thread by setting the look-ahead point to be the nearest point on the new spline to the drone's position and then advance as usual. If the target marker is ever observed to be on an occupied voxel or to be unreachable, we abort the round of data collection and discard any recorded data.The control thread runs at 20 Hz with the planning thread running while the control thread is suspended.

**Chase Task.** In this task, the drone follows a target marker along a smooth spline path that is 20 to 30 meters long. Using a generate and test method, we create a path for the drone to follow by using a random walk with momentum to find knot points for fitting a spline. Only adjacent voxels that are shown unoccupied in the voxel occupancy cache and which have not been previously visited

can be extended to during the random walk. If the path visits fewer than 20 voxels and cannot be extended to any voxel, we abort and attempt a new random walk. Since the mean free path through the environment is much larger than 30 meters, this is a relatively efficient way to generate paths, and no more than 10 attempts were ever required to generate a valid path.

A spline is interpolated between the voxels on the path as in the Navigation to Static Target task. During data collection, we follow this spline using a pure pursuit controller and placing a red cube on the look-ahead point. When flying using a trained model, we place a red cube on the position a look ahead point would be if we were using a pure pursuit controller. Unlike the Simple Navigation Task, the path is not updated as new occupancies are observed. Instead, if a voxel in the path is observed to be occupied, then we abort and discard this round of data collection.

---

**Algorithm S3** Chase Task

---
    **Input:** set of occupied voxels, $O$; number of steps in generated path, $N_p$
    $p = [\text{DRONEPOSITION}()]$
    **for** $i = 2; i < N_p; i ++$ **do**
        **while** $x \in O$ **do**
            $x = \text{RANDOMSTEPFROM}(p[i-1])$
        **end while**
        $p[i] = x$
    **end for**
    $s = \text{CUBICSPLINE}(p)$
    **while** $\text{PUSUITPOINT}(s)$ **do**
        $\text{MOVETO}(\text{PURSUITPOINT}(s))$
    **end while**

---

**Hiking Task.** In this task, the drone follows multiple target markers, which are placed on the surface of obstacles within the environment. Each of these target markers is at least 10 meters distance from each other target marker, within 10 to 30 meters from the ground, and constrained such that each successive target marker is in the half-space on the more distant side of the plane which the vector between the drone's starting position and the previous target marker is normal to. The target markers are placed by exhaustively checking each occupied voxel until one is found satisfying the conditions above. If no valid set of voxels are found, then we abort. During data collection, aborting for the reason happened fewer than 1 in 10 times.

Between rounds of data collection, we move the drone to a random unoccupied position to prevent similar selections of target markers and prevent being stuck in positions that had no possible set of valid target markers. For data collection, once target markers are found, we place a red cube on each target marker's position and perform the Navigate to Static Target task to reach that marker. After moving to this first target marker, we complete the subtask and perform the Navigate to Static Target task again on the next marker until every marker has been reached.

---

**Algorithm S4** PLANPATH($x$)

---
    **Input:** endpoint, $x \in \mathbb{R}^3$
    $k = \text{GREEDYSEARCH}(\text{DRONEPOSITION}, x, O)$
    $s = \text{CUBICSPLINE}(k)$
    **return** $s$

---

**Algorithm S5** Hiking Task

---
    $B = \text{GETBLAZES}()$
    **for** $b_i$ **in** B **do**
        $\text{NAVIGATETOENDPOINT}(b_i)$.
    **end for**

---

**Algorithm S6** GetBlazes

  **for** $i = 1$ **to** $N_B$ **do**
    **for** $v$ **in** SORTEDBYDISTANCE($O$) **do**
      **if** $\neg(Z_{min} < v[3] < Z_{max})$ **then**
        **continue**
      **end if**
      **if** $\neg(\forall b \in B, ||v - b|| > D_{min})$ **then**
        **continue**
      **end if**
      p = DRONEPOSITION()
      **if** $(v - p) \cdot (B[-1] - p) > ||B[-1] - p||^2$ **then**
        **continue**
      **end if**
      B[i] = $v$
      **break**
    **end for**
  **end for**

## S3.2 Network Architectures

We used the same convolutional head for all RNNs for ensuring a fair comparison.

Table S1: **Network size comparison**

| Model | Conv layers Param | RNN neurons |
|---|---|---|
| CT-GRU | 16,168 | 32 |
| ODE-RNN | 16,168 | 32 |
| LSTM | 16,168 | 32 |
| **NCP** | 16,168 | 32 |

Table S2: **Convolutional head**

| Layer | Filters | Kernel size | Strides |
|---|---|---|---|
| 1 | 16 | 5 | 3 |
| 2 | 32 | 3 | 2 |
| 3 | 64 | 2 | 2 |
| 4 | 8 | 2 | 2 |

## S3.3 Training Pipeline and Parameters

Before training, the recorded images and odometry data are loaded from file. From each training run, we select a continuous sequence of 65 records, discarding the run if less than 65 records are available. For the $i$th image for $i \in [1, 64]$, we compute the drone's displacement vector between frames $p_{i+1} - p_i$ and normalize it to produce a unit direction vector. This vector, as well as the image and gps direction vectors, are written to file.

Table S3: **Models' training hyperparameters**

| Variable | Value | Comment |
|---|---|---|
| Input resolution | 256-by-256 | Pixels |
| Input channels | 3 | 8-bit RGB color space |
| Learning rate | $5 \cdot 10^{-4}$ | ODE-RNN, LSTM, CT-GRU |
| Learning-rate | $5 \cdot 10^{-4}$ | NCP (RNN compartment) |
| Learning-rate | $5 \cdot 10^{-4}$ | NCP (convolutional head) |
| $\beta_1$ | 0.9 | Parameter of Adam |
| $\beta_2$ | 0.999 | Parameter of Adam |
| Minibatch-size | 8 | |
| Training sequences length | 64 | time-steps |
| Max. number of training epochs | 30 | Validation set determines actual epochs |

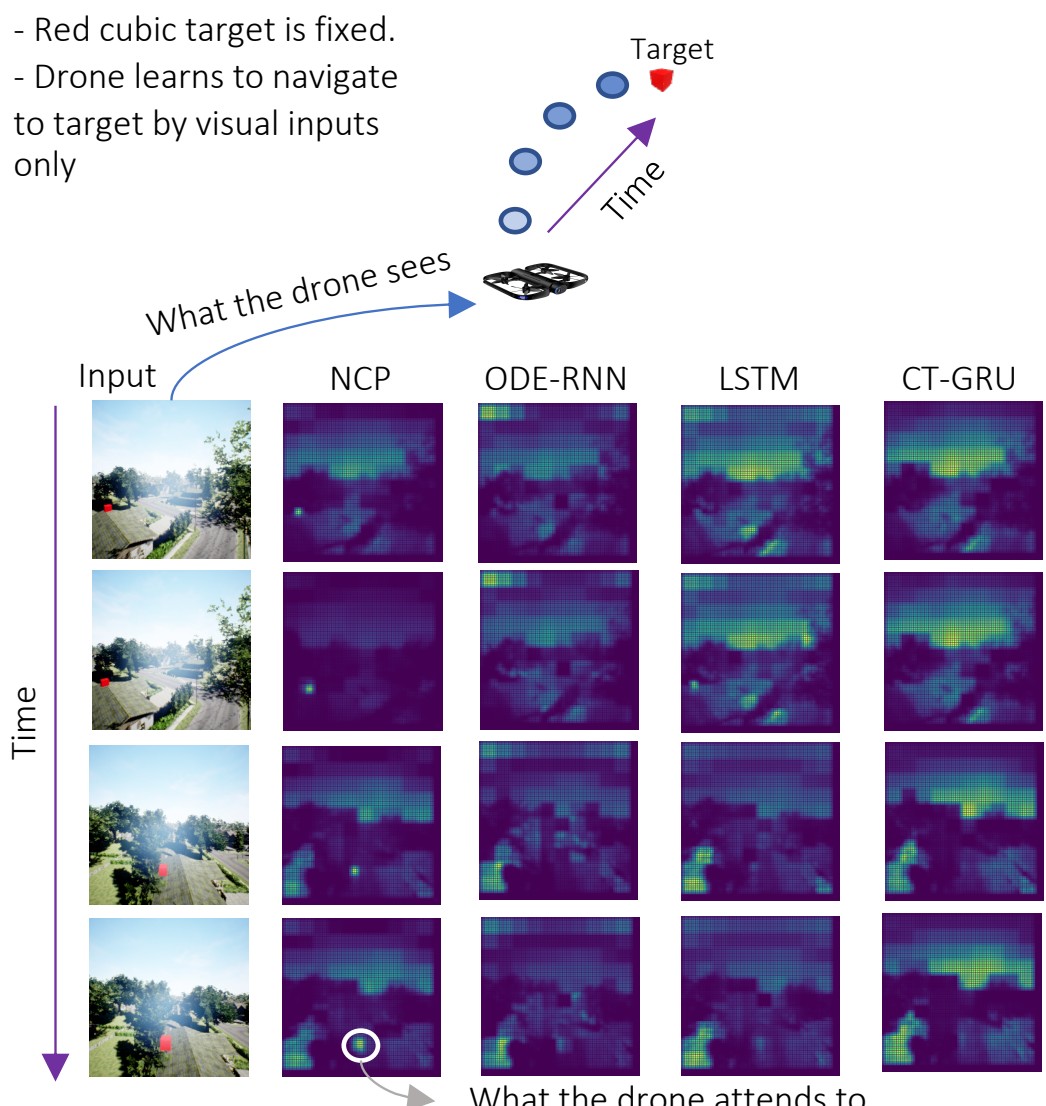

Figure S1: **Causal navigation from raw visual inputs**. Given a sequence of raw RGB inputs (left) a drone is trained to navigate towards the red-cube target. We visualize the instantaneous saliency maps (right) for each model. This paper investigates the ability of neural circuit policies Lechner et al. (2020a) (a specific representation of CT neural networks) to learn causal relationships (i.e., attend specifically to the target red-cube) directly from data while other models fail to do so. ODE-RNNs Rubanova et al. (2019b), LSTM Hochreiter and Schmidhuber (1997) and CT- Gated Recurrent Units Mozer et al. (2017). Saliency maps are computed by the visual backprop algorithm Bojarski et al. (2016).

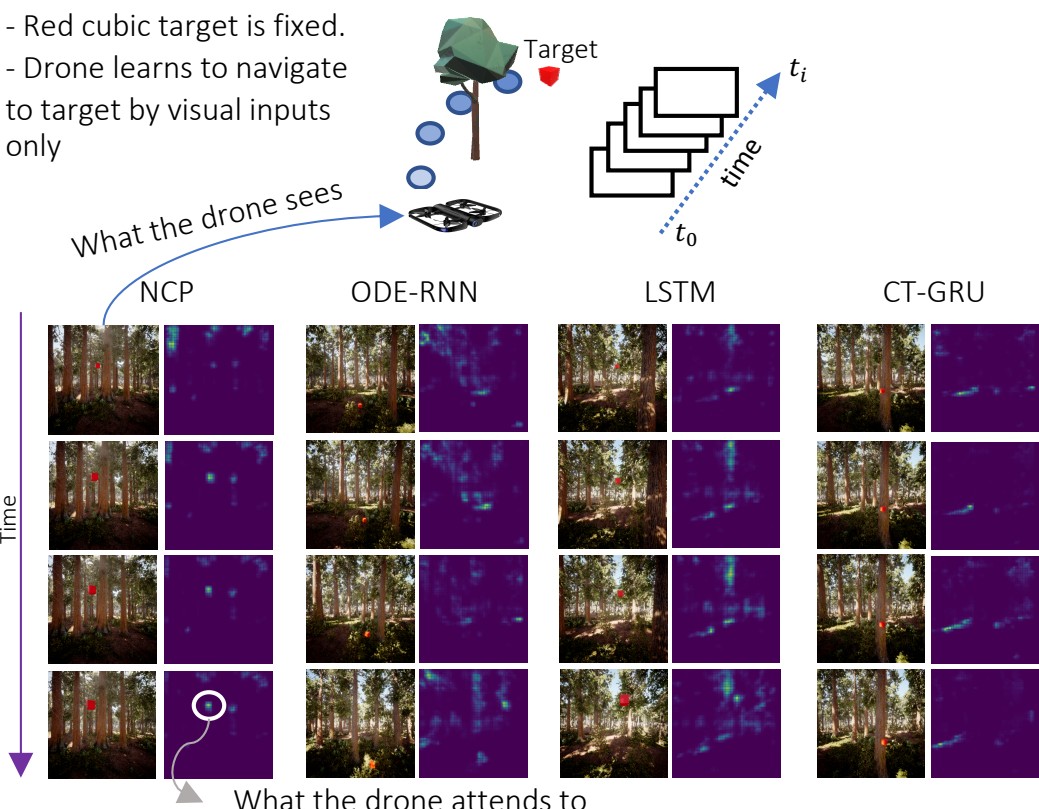

- Red cubic target is fixed.
- Drone learns to navigate to target by visual inputs only

What the drone sees

Target

$t_i$

time

$t_0$

NCP    ODE-RNN    LSTM    CT-GRU

Time

What the drone attends to

Figure S2: Navigation to a static target in closed-loop environments. NCPs are the only models that can capture the causal structure of the tasks directly from visual data.

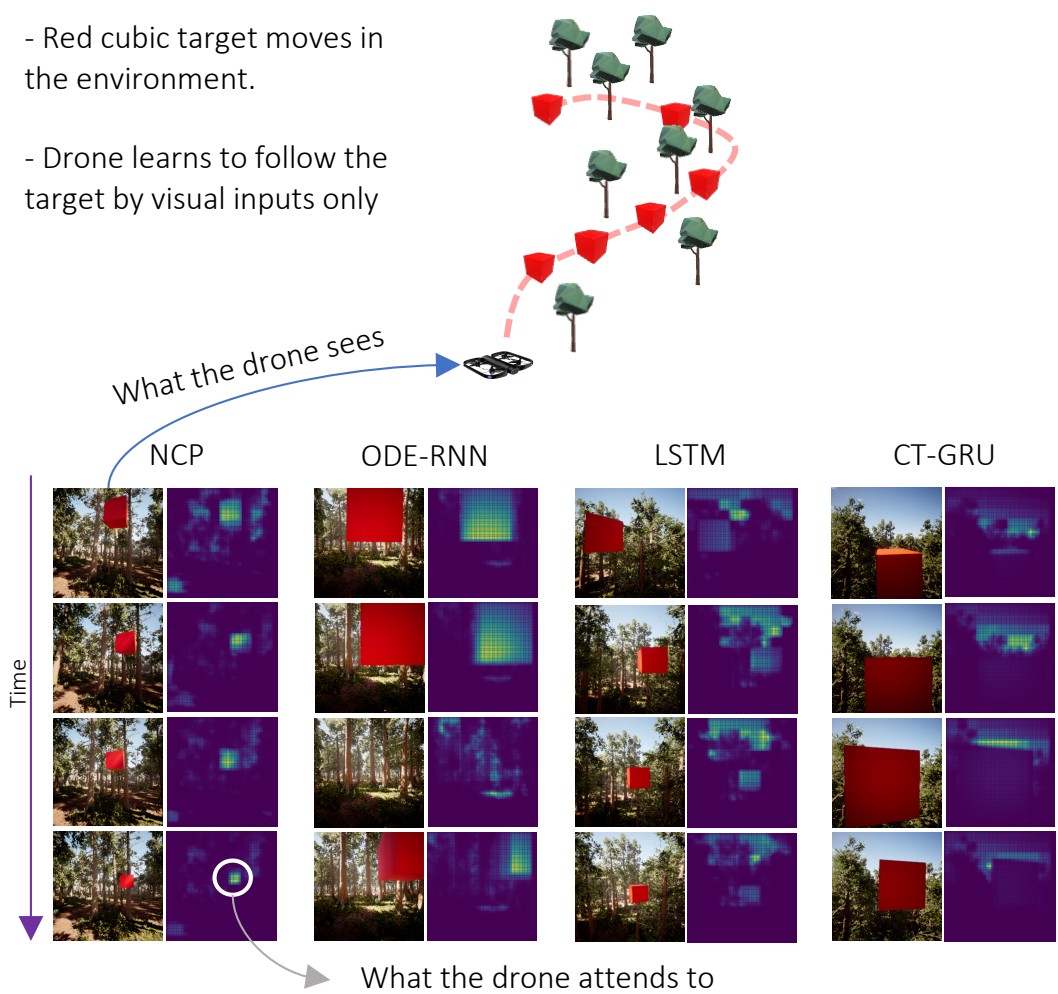

- Red cubic target moves in the environment.

- Drone learns to follow the target by visual inputs only

What the drone sees

NCP  ODE-RNN  LSTM  CT-GRU

Time

What the drone attends to

Figure S3: Chasing a moving target in closed-loop environments. NCPs are the only models that can capture the causal structure of the tasks directly from visual data.