# OpenReview forum: "Causal Navigation by Continuous-time Neural Networks"
_NeurIPS.cc/2021/Conference — NeurIPS 2021 Poster_

### Official Review · Reviewer_cZAE · 2021-07-12

**Rating:** 7
**Confidence:** 3

**Summary:**

This paper explores the capabilities of continuous-time networks to form casualty structures, as well as their applications to visual drone navigation. The authors find that models based on the liquid time-constant networks (LTCs) uniquely provide causal structures. The theoretical discoveries are then verified in closed-loop visual navigation experiments. LTC-based policies significantly outperform regular continuous time models and discrete RNNs, all trained using imitation learning.


**Limitations And Societal Impact:**

Discussed in the paper.

**Main Review:**

=== Strengths ===

+ This paper provides theoretical analysis on whether various continuous-time networks can form causal structures. The authors show that LTCs are causal models whereas other continuous models such as regular neural ODEs are not.

+ LTC-based policies significantly outperform baselines in closed-loop experiments. This shows great potential of LTC-based models being a generic policy architecture in robotics or RL.

+ I appreciate the saliency analysis across models as it futhers shows the causal structure captured by the training data

=== Weaknesses ===

- Experiments in table 4 shows LTC’s effectiveness as it outperforms LSTM as a discrete RNN baseline, and ODE-RNN as a CT baseline. I am curious to see a vanilla CNN baseline and how it performs in the same tasks, as prior work has shown regular RNNs can suffer from the causal confusion problem in the imitation learning setting [1,2].

=== Other comments ===

- LTC (or NCP) as a generic policy architecture provides causal structures as the authors show in the imitation learning experiments. I am curious how it would perform against other architectures in the RL setting.`

\
[1] Casual Confusion in Imitation Learning, de Haan et al., 2019

[2] Fighting Copycat Agents in Behavioral Cloning from Observation Histories, Wen et al., 2020


**Time Spent Reviewing:**

5

---

> ### Author Response · Authors · 2021-08-10
> **Response to Reviewer cZAE**
>
> Thank you for the positive evaluation of our work and your constructive feedback on our manuscript. Please find our response to your concerns in the following:
>
> **How would a CNN network work in these scenarios?**  As requested by the reviewer, we performed an additional experiment in all cases with CNNs. In the table below we can see the success rate of CNNs in all tasks when deployed in closed-loop, including the robustness analysis, and compared to the NCPs. As expected, we observe that when temporal dependencies in the tasks appear (such as Occlusion, or Hiking) as well as when the input images are highly perturbed (such as heavy rain and Fog) the performance of the CNN is drastically decreased.
>
> | | |Static Target |  | | | | Chasing| | | Hiking |
> |---|---|---|---|---|---|---| ---| ---| ---| ---|
> |  Model | Clear | Fog | Light Rain | Heavy Rain | Occlusion | Clear | Fog | Light Rain | Heavy Rain | Clear |
> | CNN | 36% | 6% | 32% | 2% |  4%  | 50% |  42%  | 54% |  28% | 0% |
> | NCP (ours) | 48% | 40% | 52% | 60% | 32% | 78%|  52% | 84% | 54% | 30% |
>
> **NCPs in RL?**  This is an exciting line of work that we have down the line. Our early results on training NCP policies for single-agent flight control tasks (such as drone and fixed-wing vehicle dodgeball) and multi-agent tasks (environment surveillance with multiple drone agents with a collaborative objective), show great promise. Compact NCP agents can be used in both model-free and model-based RL schemes. Additionally, [Hasani et al. ICML 2020], studied LTC-based networks in the context of simple search-based RL setting:
> Hasani, R., Lechner, M., Amini, A., Rus, D., & Grosu, R. (2020, November). A Natural Lottery Ticket Winner: Reinforcement Learning with Ordinary Neural Circuits. In International Conference on Machine Learning (pp. 4082-4093). PMLR.

---

### Official Review · Reviewer_JrdD · 2021-07-16

**Rating:** 5
**Confidence:** 4

**Summary:**

The manuscript studied a formulation of continuous-time neural networks and discussed its ability to learn causal structures in the environment.

The manuscript made claims on the following contributions:
- Providing theoretical background and insights on how continuous-time neural networks learn causal structures
- Experiments on visual drone navigation tasks
- Analysis of robustness in closed-loop control settings

**Limitations And Societal Impact:**

Yes

**Main Review:**

Section 2, Re: Imitation, Visual Navigation: There is some missing discussion here, in the context of visuomotor control for navigation. The learning-to-drive community is actively attempting to address issues in imitation learning, such as causal confusion [10] and generalisability, through attempting to approximate informative priors [DATF, DIM, RIP], coupling policies with classical controllers [DIM, RIP], knowledge distillation [LBC], affordance learning [CAL], and others. It would be more appropriate for the present manuscript to contextualise its contributions with respect to these advances.

L65, 256-257: The manuscript makes a general assertion that 'learning from offline demonstrations (for deployment to online settings)' (let's call this condition #1) is easier than 'learning from online observations and feedback (for deployment to online settings)' (condition #2) and, implicitly, 'learning from offline demonstrations, online observations, and online feedback (for deployment to online settings)' (condition #3). This is misleading and, relative to the justification provided in the manuscript, fundamentally imprecise; perhaps more realistic scenarios, such as those mentioned in my other comments should be considered by the manuscript. A major issue with i.i.d. imitation (besides limited behaviour/event mode coverage, from distribution shift and mismatched support) is that there is limited or non-existent observability over the environment and system dynamics that might result in particular outcomes. Furthermore, it is more difficult to resolve multi-agent causal relationships (e.g., traffic light causes pedestrian to walk, traffic light causes car to stop, pedestrian causes car to stop, car causes pedestrian to stop, etc.) in an imitative learning paradigm. In these settings, online optimisation (where characterisation of environment and/or system dynamics is available) can actually make for an easier objective.

Section 4.1: The observation that continuous-time inductive biases provide models with robustness to causal confusion and distribution shift is not new, nor are couplings between structured physical models and neural systems. Both have been incorporated in other approaches for visual navigation and trajectory forecasting. See [DATF] (re: Verlet Integration), [DIM], and [RIP].

L184, 185: "Results", "results" --> "Approach", "approach"

Section 5: While additional tasks are always nice, I would like to see comparison of the work presented in this manuscript with other approaches on more well-established benchmarks. This is more than just a request to perform more experiments: the precise problems that this manuscript is attempting to alleviate (causal confusion, generalisability) are exactly the challenges that such tasks as CARLA NoCrash, CARNOVEL, and CARLA Challenge are designed to provide assessments for. Moreover, there already exist several contemporary works that have used these benchmarks to study these problems, thereby eliminating the need to only use simple RNN-class baselines.

L42: Missing word: "(i.i.d) (see Fig. 2)" --> "(i.i.d) observations (see Fig. 2)"

--------

References

[DATF] Park, Seong Hyeon, Gyubok Lee, Jimin Seo, Manoj Bhat, Minseok Kang, Jonathan Francis, Ashwin Jadhav, Paul Pu Liang, and Louis-Philippe Morency. "Diverse and admissible trajectory forecasting through multimodal context understanding." In European Conference on Computer Vision, pp. 282-298. Springer, Cham, 2020.

[DIM] Rhinehart, Nicholas, Rowan McAllister, and Sergey Levine. "Deep Imitative Models for Flexible Inference, Planning, and Control." In International Conference on Learning Representations. 2019.

[RIP] Filos, Angelos, Panagiotis Tigkas, Rowan McAllister, Nicholas Rhinehart, Sergey Levine, and Yarin Gal. "Can autonomous vehicles identify, recover from, and adapt to distribution shifts?." In International Conference on Machine Learning, pp. 3145-3153. PMLR, 2020.

[LBC] Chen, Dian, Brady Zhou, Vladlen Koltun, and Philipp Krähenbühl. "Learning by cheating." In Conference on Robot Learning, pp. 66-75. PMLR, 2020.

[CAL] Sauer, Axel, Nikolay Savinov, and Andreas Geiger. "Conditional affordance learning for driving in urban environments." In Conference on Robot Learning, pp. 237-252. PMLR, 2018.

**Time Spent Reviewing:**

4

---

> ### Author Response · Authors · 2021-08-10
> **Response to Reviewer JrdD**
>
> Thanks for your feedback on our manuscript. Before getting into a point-by-point response, we would like to emphasize that the proposed contribution of our work was not to solve the causal confusion problem of the RNNs nor their generalization in imitation learning. Rather we investigate the capability of continuous time models in learning causal control policies for online, closed-loop testing using only offline, passive expert trajectories.
>
> We fundamentally showed theoretically and experimentally how neural networks built by the liquid time-constant (LTC) principle, as a subclass of continuous-time (CT) networks (an emerging class of neural network models), are dynamic causal models and without any regularization, can learn the causal structure of the visual task they are given.
>
> We greatly appreciate the reviewer bringing these related works to our attention. We will include discussion of all in a potential camera-ready version of our work. Furthermore, we additionally discuss the relevance of these works in the context of our contribution below:
>
> [DATF]: This paper presents an attention-based method to extract context from multimodal inputs for performing “diverse” and “admissible” trajectory predictions. The paper shows promising results in doing so in a set of multi-agent autonomous driving scenarios. Our paper proves that LTC-based models, as a subclass of CT networks, are dynamic causal models and can capture the true causal structure of the task they are given. Although the method proposed in [DATF] extracts useful features for performing trajectory prediction by using the “cross-agent attention” blocks, the interpretation of the extracted (learned) concepts and features has not been discussed and is challenging to obtain due to the parameter overhead. In contrast, this interpretation is extensively discussed and quantitatively evaluated on NCPs. In fact, NCPs are interpretable up to their cell level due to the use of significantly less number of trainable parameters (3-4 orders of magnitude fewer parameters than networks based on attention) see for example:
> Lechner, M., Hasani, R., Amini, A., Henzinger, T. A., Rus, D., & Grosu, R. (2020). Neural circuit policies enabling auditable autonomy. Nature Machine Intelligence, 2(10), 642-652.
>
> Additionally, we do see a complementary aspect in our work compared to [DATF] which could further improve the impact of our results. The cross-agent attention encoder can be combined with NCPs instead of the LSTMs and the variants used as baseline to potentially gain better performances, based on what our paper presented.
>
> [DIM]: This paper presents a multi-step imitation learning algorithm for goal-planning scenarios under different objectives. The paper is a great example of how to achieve better performance in IL. An imitative model is trained over human data, and then used as an imitative planner at the test time. It receives input waypoints from a route planner, and delivers its path predictions to a PID controller to perform human-like decisions. Again, in our present manuscript, our findings are fully complementary to that of [DIM]. In particular, we can design better imitative models for instance by replacing the RNN modules of Fig. 4 of [DIM], with NCPs to gain better insights about the dynamics of the “imitative agent”.
>
> [RIP] This paper’s objective is not directly relevant to our main goal as the method proposed tries to tackle distribution shifts and dealing with out-of-distribution scenarios with an uncertainty aware method. The models presented in [RIP], however, consist of two heads synapsing into an RNN. The RNN can be chosen to be an NCP network as our results here present consistently better performance achieved by NCPs compared to advanced RNN baselines.
>
> [LBC] This work presents an effective and very clever method for training better IL agents for vision-based autonomous driving. A ResNet-18 backbone + heads is first trained by privileged information and a ResNet-32 is used as a sensorimotor backbone for the second stage of training without privileged information. The fundamental advantage of NCPs compared to the networks used in [LBC] is that NCPs realize networks with 2-4 orders of magnitude less trainable parameters than the networks used here for the same task (see the size of the networks presented in this work in the supplementary materials + see an example of NCPs for lane-keeping here: Lechner, M., Hasani, R., Amini, A., Henzinger, T. A., Rus, D., & Grosu, R. (2020). Neural circuit policies enabling auditable autonomy. Nature Machine Intelligence, 2(10), 642-652.)
>
> This reduction in size and NCPs’ semantics allows the network to be interpretable at the cell level. A property not achievable by any of the networks used in the suggested related works. Moreover, the ability of NCPs to distill causal representations with their extremely compact networks, alone, is beyond what is achievable with state-of-the-art RNN baselines as demonstrated in our work.
>
>
> [CAL]: In this paper, LSTMs, GRU, and temporal convolutions are used as task-specific heads. Experiments with NCPs as heads could potentially improve the performance and the interpretability of the models for autonomous driving.
>
> From this set of related works and the comments of the reviewer, we see a great path towards writing up a new paper on deploying NCPs for self-driving cars that shows how NCPs can be a proper replacement for the model architecture in various IL settings. We believe that our present study
> (1) formulates the foundation for theoretically and experimentally validating the ability of the LTC-based models to learn causal dynamics; and
> (2) is a complete piece of work that can lead to many potential applications to bridge this effect on other sensorimotor tasks, such as autonomous driving in CARLA, with various imitation learning algorithms + NCPs.
>
> L65, L256-L257: We fully agree with the reviewer, and by the sentences in lines 65, and 256-257 we did not mean to suggest that there is a contest between “condition 1” “2” and “3”. We will clarify this further using the comments of the reviewer, in our revised manuscript to avoid misunderstandings. Please denote that we did not aim to provide the ultimate solution or even a solution to the IL problem. We showcased how causality can be achieved for the class of CT models by LTCs, and exemplified our theoretical results with drone visual navigation tasks with increasing complexity.
>
>
> Section 4.1: We would like to clarify that we are not claiming to solve the causal confusion task in this work. Even so, we would greatly appreciate it if the reviewer can mention a previous work that shows CT models are less sensitive to causal confusion. In our work, we are showing that Liquid networks are fundamentally dynamic causal models which can lead to the increased active testing performance of IL models (to the best of our knowledge, this has not been shown elsewhere).
>
> Moreover, the connection between the structured physical models and neural systems is a fundamental property of any time-continuous neural system (not only LTCs) - and we refrain from claiming this as a novel contribution of our work. In fact, there are other CT models which we also benchmark against in our experimental evaluation to solidify this comparison. What is new in our work is to show a great match between our theoretical results on the causality of LTC-based models in a simulated real-world task by qualitative inspection of the saliency profile of the networks. And that in our diverse set of experiments (and the robustness analysis), these networks are more performant, more interpretable, and more capable of understanding the task they are given to perform.
>
>
> Section 5: We fully agree with the reviewer’s suggested type of analysis and that this is beyond adding additional experiments to the paper. It is indeed a new paper on imitation learning for autonomous driving with NCPs. This is great advice which we will take action on in our continued effort.
>
> We hope to have addressed the concerns of the reviewer and are looking forward to receiving their response to our rebuttal.

---

### Official Review · Reviewer_G1FC · 2021-07-19

**Rating:** 7
**Confidence:** 3

**Summary:**

This submission proposes a theoretical and experimental framework for learning causal representations using continuous-time neural networks. The proposed method is evaluated on a visual drone navigation tasks of chasing static and dynamic objects in the AirSim simulation. Experiments show that the proposed method outperforms LSTMs, recurrent neural network version of Neural ODEs and continuous-time GRUs.


**Limitations And Societal Impact:**

Yes

**Main Review:**

Strengths:

- The paper is well motivated. Learning causal relationships is an important machine learning problem. Learning visual navigation policies is a good testbed for this problem.

- The paper is written very well. It is easy to follow and the method is described well.

- The method is novel to the best of my knowledge. It leverages continuous-time neural networks to learn casual relationships using a class of liquid time-constant networks.

- The experiments is well designed. The authors test the method on a series of visual drone navigation tasks with increasing difficulty and implement a good variety of baselines.

- The experiments show the effectiveness of the method empirically. The attention visualization is also informative. Supplementary material provides good video visualizations

- The supplementary material also provides code for reproducibility.


Weaknesses:

- The proposed method performs worse than the baselines for a few settings. The submission will benefit from some experiments for a different task in a different simulator to establish the consistency of results.

- It would be good to add map-based methods mentioned in the visual navigation paragraph in the related work as baselines. Some of these methods such as CMP or Neural SLAM are shown to perform much better than LSTM policies.

- All the methods perform poorly on the Hiking task. It would be good to discuss the limitations of NCP and author's thoughts on what needs to be improved to tackle complex tasks such as Hiking.

**Time Spent Reviewing:**

2

---

> ### Author Response · Authors · 2021-08-10
> **Response to Reviewer G1FC**
>
> We would like to thank you for the positive evaluation of our work and your constructive feedback on our manuscript.
>
> **Weaknesses Addressed**
>
> *Performance on baselines:*
>
> As it was well-mentioned by the reviewer, our objective in this work was not to get a model with the highest offline performance. In contrast, we tried to test the ability of the models to learn causal structures and their capability of understanding the task they are performing in an online, closed-loop test scenario, under various challenging environmental disturbances.
>
> We confirmed that in all tasks, NCPs are the only models that can capture the true causal structure of the task under test (see the saliency attention maps). Moreover, apart from the “Light Rain” environment (in the static target following), and the “Foggy” environment (in the chasing task), in all remaining tasks NCPs significantly quantitatively outperform the other baselines in terms of their success rate in online testing.
>
> *On map-based methods:*
>
> We fully agree with the reviewer that map-based baselines can perform these tasks very well. However, our objective here was to show the causal properties of recurrent models which separates our experimental setting from that of map-based methods. Furthermore, recurrent end-to-end models have a distinct advantage over map-based baselines in which they do not require collecting any data from the test environment before deployment (train and test scenes are separate).
>
> *How to tackle challenging tasks such as hiking:*
>
> This is an excellent suggestion which we will add to our revised manuscript: In order to improve the performance on a purely visually-navigated hiking task, an agent must be supplied with a long-term memory component (much longer than that of LSTMs). All ODE-based models including NCPs require a gradient wrapper to be able to perform well on tasks with very long-term dependencies. For instance, mixed memory architectures such as ODE-LSTM [Lechner and Hasani et al. 2020] can use NCPs as their continuous-time memory wrapped together with gradient propagation mechanisms enabled by an LSTM network, to perform better on the hiking task.
>
> We hope to have addressed the comments of the reviewer and will make sure to incorporate them properly in a revised version of our manuscript.

---

### Decision · Program_Chairs · 2021-09-27

**Decision:**

Accept (Poster)

**Comment:**

This paper introduces a method for visual navigation using continuous time (CT) neural networks in the continuous liquid time neural networks (LTC). It claims, that this variant is able to learn causal structures, compared to baseline CT models as well as standard deep models like LSTMs, and evaluates these claims on drone navigation tasks.

The paper received 3 expert reviews, which were on the fence. After the discussion phase, two of the reviewers were clearly positive of the paper, and the remaining reviewers was borderline slightly negative.

Several weaknesses were pointed out by the critical reviewer:
- the limitations of positioning as imitation learning
- novelty of the claim that continuous-time formulations increase robustness wrt to causal confusion
- Simplicity of the tasks

The authors could provide answers to most of these issues, but the remaining weakness was the simplicity of the experiments.

The AC's reading of the paper was positive. He agrees on the simplicity of the tasks, but judges that this paper has sufficient merits and novelty to be of interest to the community.

The paper was then discussed between AC and SAC, who confirms the decision.